# Types of Client Perceptions Regarding Counselling Experiences at Psychological Counselling Centres: Utilising Q Methodology

**DOI:** 10.3390/bs14070586

**Published:** 2024-07-10

**Authors:** Kyoung Hwa Yi, Yeon ah Lim, Jeeyoung Lee, Song Yi Lee

**Affiliations:** 1Coaching and Counselling, Graduate Schools, Dongguk University-Seoul, 30, Pildong-ro 1 gil, Jung-gu, Seoul 04620, Republic of Korea; newself90@dgu.ac.kr (K.H.Y.); 2022126711@dug.kr (Y.a.L.); 2021120400@dgu.ac.kr (J.L.); 2Dharma College, Dongguk University-Seoul, 30, Pildong-ro 1 gil, Jung-gu, Seoul 04620, Republic of Korea

**Keywords:** psychological counselling centre, counselling experience, client, Q methodology

## Abstract

This study used Q methodology to explore the various types and characteristics of clients’ subjective perceptions concerning their experiences at psychological counselling centres. We selected 33 Q samples from a Q population of 135; of the Q sample, 31 P samples underwent Q sorting. Subsequently, we analysed the data using the Quanl Program. The study categorised perception into four distinct types. Type 1 values therapeutic counselling relationships, Type 2 prioritises counselling services, Type 3 values counsellor assignment, and Type 4 prioritises the counselling structure. This study provides valuable basic data to clients, counsellors, and counselling institutions.

## 1. Introduction

Humans possess emotional functions that machines, such as robots and artificial intelligence, lack. Thus, they manifest psychological characteristics such as empathy ability and emotional expression [1]. These emotional functions can cause humans to experience psychological difficulties and loneliness [2]. In the past, Korea’s analogue culture fostered the development of close-knit communities and extended family structures. This environment offered a context in which people could seek psychological comfort through deep conversations with family members and friends [3]. However, as individualistic tendencies strengthen with the advancement of industrialisation and modernisation, the community culture is becoming increasingly rare [4].

The onset of coronavirus pandemic in 2019 expedited these changes even more. As individuals’ isolation deepened and forming social relationships became more complex, they experienced a growing level of stress and emotional deficiency [5]. Especially during the COVID-19 pandemic’s isolation experience, video call technology contributed significantly to maintaining social relationships and pursuing emotional stability [6]. In addition, as social interactions diminished during this period, people experienced an augmented inclination to alleviate their psychological pain by engaging in conversations with others [7]. This factor further enhanced people’s motivation to address their internal problems through psychological counselling [8].

To tackle internal issues, one must understand human behaviour and thought processes, which form the foundation for psychology [9]. Counselling psychology is a specialised field that focuses on addressing psychological problems and supporting clients through counselling relationships. This field encompasses research on counselling theory and practice and provides counsellors with the necessary knowledge and skills [10]. The therapeutic relationship between a counsellor and a client is important in resolving clients’ maladaptive behaviours and psychological difficulties [11]. Through their relationship, the counsellor and client establish explicit goals to address the client’s major concerns. They collaborate to achieve these goals, ensuring that they align with the client’s agreement throughout the counselling process [12]. Appropriate counsellor assignments are important in counselling institutions. It increases counselling effects and early termination or minimising counselling complaints [13]. The bond and rapport between counsellors and clients are critical to counselling success [14].

Clients should be aware of the criteria for choosing a counsellor and their options at a counselling institution. Conventionally, counsellors have provided psychological counselling services by having clients visit their offices. During these visits, counsellors would explain the process briefly, followed by counselling sessions [15]. When psychological counselling commences, the counsellor establishes the structure and explains the process and the ethics regulations. In this process, clients encounter challenges in identifying their counsellor preference and selecting a counsellor. Moreover, novice or trained counsellors find it challenging to effectively understand the client. Therefore, a more effective method is necessary to maximize the counselling effect and meet the client’s expectations.

Previous studies on clients’ counselling experiences are crucial for understanding the effectiveness and significance of psychological counselling. Multiple domestic and international studies have explored the impact of clients’ counselling experiences on improving their psychological stability and overall quality of life. For example, a domestic study investigated the relationship between adolescents’ cyber counselling experiences and professional help-seeking attitudes. It demonstrated the influence of clients’ counselling experiences on their help-seeking behaviours [3]. Adolescent cyber counselling refers to counselling services that are provided to adolescents through the internet, utilising various online platforms such as chat, email, and video calls. This online counselling form includes real-time, text-based communication between the client and the counsellor. Typically, it is conducted via the internet or mobile applications, thereby offering a convenient and accessible form of support. In addition, a study researched the perceptions of adolescent clients and counsellors concerning their online counselling experiences. It discovered that counselling experiences are essential in forming relationships with counsellors [4].

An overseas study analysed clients’ various counselling experiences. It emphasised the correlation between the quality of counselling and client satisfaction [16]. Furthermore, another study highlighted the significance of counselling experiences by researching how clients develop self-understanding and acceptance through counselling [17]. Such studies demonstrate that a client’s counselling experience impacts psychological problem-solving and emotional stability.

Examining clients’ subjective perceptions is vital for maximising the effect of psychological counselling. Due to diverse backgrounds and experiences of clients, their counselling expectations and needs vary. Therefore, comprehending clients’ subjective perceptions is essential for counsellors to select appropriate counselling approaches and provide more personalised counselling services.

By understanding various subjective perceptions, counsellors can establish trusting relationships with clients, set counselling goals effectively, and meet clients’ needs during the counselling process. For example, clients who prioritise the therapeutic counselling relationship seek to enhance their self-understanding and growth by forming a trusting relationship with the counsellor. In contrast, clients who prioritise the quality of counselling services regard friendly counsellors and confidentiality as important. Understanding and reflecting on these differences is essential to enhance the effects of counselling. This is corroborated by various studies, including Timulak and Keogh (2017) [17], who emphasise the importance of self-understanding and acceptance in counselling, and Anderson et al. (2009) [18], who highlight the role of counsellor friendliness and confidentiality in client satisfaction.

In addition, examining subjective perceptions can enhance the ability of counselling institutions’ to provide superior client services by understanding the various factors that clients consider when choosing a counselling centre. This strategy enhances clients’ revisit intentions and helps improve the counselling centre’s reputation. ‘Clients’ revisit intentions’ refer to the likelihood or inclination of clients to return to a counselling centre for further sessions following their initial visit. Clients’ level of satisfaction with the counselling experience, their rapport with the counsellor, and the effectiveness of the counselling are the factors that determine this intention to revisit [19,20].

Presenting previous studies on clients’ counselling experiences and analysing various subjective perceptions can increase the effectiveness of psychological counselling, resulting in benefits for both counsellors and clients. Although it is crucial to understand the subjective perception and emotional experience of clients during psychological counselling experiences, such studies are rare. This study employs Q methodology to investigate the types of client perceptions of their counselling experiences at psychological counselling centres. The study’s results offer basic data for clients to select a suitable counsellor and counselling institution. Therefore, this study’s research questions are:
RQ1. What are the types of perceptions clients have regarding their counselling experiences at psychological counselling centres?RQ2. What are the distinguishing characteristics of each type of perception identified?

## 2. Study Design

### Study Method

This research used Q methodology, a research approach that specifically explore clients’ subjectivity. Q methodology helps researchers rediscover counsellors’ and clients’ diverse thoughts, opinions, values, and choices within counselling relationships in factual counselling contexts [21]. Most social science studies use quantified data to explain social phenomena and behaviours; they frequently exclude the subjectivity of researchers and study participants [22]. However, individuals’ perception of most social phenomena can vary depending on their concerns and interests.

In particular, psychological counselling experiences occur within individuals’ irregular and uncertain realities and are characterised by strong subjectivity [23]. Thus, the Q methodology is useful for confirming subjectivity and investigating responses to certain topics or event stimuli [24]. This methodology is a quantitative research technique because it utilises statistical factor analysis. Moreover, it is a qualitative research method because it involves exploring subjectivity through the self-reference of study participants [25].

Therefore, this study aims to investigate the subjective perception and evaluation of clients regarding their counselling experiences at psychological counselling centres. The different perceptions derived through the Q methodology reflect clients’ individual experiences and subjectivity. Based on this, we can derive the effect of psychological counselling and improvement plans. This study used the following key steps: The first step entailed constructing the Q population. We collected 384 statements through in-depth interviews and literature reviews. Subsequently, they were refined to 135 statements. The second step involved the selection of Q samples. Based on representativeness and comprehensiveness, 33 Q samples were selected from the Q population. The third step entailed the selection of the P sample. Thirty-one participants were selected to represent clients’ diverse experiences, including various ages and genders. The fourth step involved the process of Q sorting. The P sample participants sorted the Q samples to convey their subjective perceptions. The final step involved data analysis. We analysed the Q sorting results to identify and interpret the major perception types. Figure 1 shows a schematic of this study’s research procedure.

## 3. Research Procedure

### 3.1. Q Population and Q Samples

A Q population is an assembly of items gathered for Q research, consisting of self-referent statements of opinions capable of projecting oneself rather than facts [26]. To that end, we collected data on counselling experiences at psychological counselling centres to form a Q population. Subsequently, we conducted in-depth interviews using semi-structured questionnaires, and further supplemented the Q population by adding additional statements. We conducted the preliminary survey to construct the Q population, consisting of study participants with counselling experiences at public institution psychological counselling centres, university institution counselling centres, and individual psychological counselling centres. This study commenced after receiving the Institutional Review Board (IRB) approval. Before participants’ engagement in the research, we explained the study’s purpose and obtained their written consent.

The in-depth interviews comprised eight semi-structured questions focused on the following: (1) the motivation for utilising the psychological counselling centre, (2) the characteristics of the psychological counselling centre, (3) the individuals’ feelings towards the psychological counselling centre, (4) the individuals’ experiences at the psychological counselling centre, (5) the aspects of counselling at the psychological counselling centre that were satisfactory, (6) the aspects that were dissatisfying, (7) whether the participant would consider returning to the psychological counselling centre and the reasons behind their decision, and (8) the reasons why individuals who had no intention of revisiting the centre made that choice.

A Q population of 384 statements was obtained from the content of these in-depth interviews and literature reviews. Through this process, 135 statements were selected from the initial pool of 384 based on representativeness and comprehensiveness. Subsequently, these 135 statements were further refined to reach a final set of 33 Q-sample statements. Throughout this process, we applied the following criteria: First, representativeness: each statement was evaluated to ascertain if it adequately reflected the experiences at the psychological counselling centre. This aimed to capture clients’ diverse experiences and emotions. Second, comprehensiveness: we assessed whether the statements sufficiently encompassed various subjective perceptions. This involved selecting statements that addressed multiple aspects of the counselling experience. Third, clarity: each statement was evaluated for clarity and specificity. This entailed ensuring that the statements were easily understandable and free from ambiguity. The final set of 33 Q-sample statements encompassed various aspects such as motivations for using the counselling centre, characteristics of the centre, counselling experiences, points of satisfaction, points of dissatisfaction, and intentions to revisit. Table 1 presents the 33 Q-sample statements. We extracted 135 statements through this process, and then selected 33 statements as Q-samples.

### 3.2. P Sample Selection

P samples refer to study participants who completed a questionnaire containing statements aimed at categorising different types. Scholars recommend selecting a small cohort of 20 to 40 participants who are relevant to the research topic [27]. Thus, after receiving IRB approval on 15 July 2022, this study focused on clients who had counselling experiences at public institution counselling centres, university institution counselling centres, or individual counselling centres over a period of two months.

First, we posted recruitment notices for clients who have experience receiving psychological counselling at public institutions, university institutions, or individual counselling centres. This notice included detailed information regarding the study’s objectives, procedures, eligibility criteria, and compensation provided to participants. Clients who wished to participate in the study through the recruitment notices completed the initial survey via an online questionnaire or telephone interview.

The initial survey gathered basic information, such as the client’s counselling experience, the type of institution where they received counselling, the counselling period, and the number of counselling sessions. We utilized the basic information to evaluate the client’s suitability to participate in the study. The criteria for assessing suitability for participation in the study were as follows: (1) those who received counselling at a psychological counselling centre within the last year, (2) those who can represent the experiences of clients, including those of various ages and genders, and (3) those who sufficiently understand the study’s purpose and procedures and intend to participate voluntarily in the study. Participants eligible for the study were defined as those who could represent clients’ experiences, which encompassed a diverse range of ages and genders. This definition ensures that the participants’ experiences adequately reflect the various experiences encountered at psychological counselling centres. For example, clients who received counselling at a specific centre can naturally represent their experiences there. However, to maintain the study’s reliability and objectivity, clients with diverse backgrounds and experiences should be incorporated.

Notably, this study included adolescents. The reason is that most studies focus exclusively on either adults or adolescents. However, this study aimed to explore the diversity of counselling experiences across various age groups. The adolescents in this study played a crucial role in enhancing our understanding of how their experiences might differ from those of other age groups. Adolescents’ counselling experiences can differ from those of adults, and analysing these differences enables a broader understanding of the effectiveness of psychological counselling.

No overlap was observed among these two groups. Because this study aims to examine perceptions of experiences at counselling centres, individuals without experience using counselling centres are excluded. Individuals with cognitive impairments that prevent them from expressing their experiences are also excluded.

Based on the criteria, we selected a P sample of 31 study participants (males, *n* = 8; females, *n* = 23). The participants exhibited an even distribution across age groups: teens (*n* = 2), twenties (*n* = 8), thirties (*n* = 4), forties (*n* = 6), fifties (*n* = 8), and sixties (*n* = 3). Once again, we explained the purpose and procedures of the study in detail to the selected study participants. We requested them to complete and sign a consent form for participation. All participants voluntarily participated in the research and freely expressed their counselling experiences and subjective perceptions during the interview.

The study participant selection process progressed fairly and systematically, encompassing the subjective perceptions of clients with varied backgrounds and experiences. This study secured basic data to analyse clients’ subjective perceptions of their counselling experiences at psychological counselling centres from various angles. Table 2 summarises the P samples’ gender and ages.

### 3.3. Q-Sorting Process

Q-sorting refers to the process in which study participants sort the Q samples, which consist of 33 statements, and assign a score to each item. This process enables study participants to express their subjective beliefs, attitudes, feelings, thoughts, opinions, and preferences [28]. The participants read, evaluate, and sort the 33 statements (i.e., the Q samples) based on their subjective interpretations. The process involved employing a method of forcefully disseminating individual items based on the data distribution diagram illustrated in Figure 2. The participants sorted the Q-statements into three levels, ranging from those they most disagreed with to neutral to those they agreed with most. Subsequently, they arranged the statements based on their degree of agreement or disagreement, referring to the data distribution diagram model (Figure 2) and providing reasons for their selection.

The sorted scores were across nine scales, depicted in Table 3. These scales ranged from those that participants most disagreed with (−4) to those with which they most agreed (+4).

After identifying the Q statement numbers recorded in the data distribution diagram, the researchers assigned scores to the statements. They started with 1 point for the statement with the most disagreement (−4), followed by 2 points (−3), 3 points (−2), 4 points (−1), 5 points (0), 6 points (+1), 7 points (+2), 8 points (+3), and 9 points for the statement with the most agreement (+4). Once the scoring was completed, we entered the scores into a computer.

### 3.4. Data Analysis

We conducted principal component factor analysis using the statistical results generated by the Quanl Program which is developed by Norman Van Tubergen in the 1960s for mainframe platforms. We calculated the data based on factors with an eigenvalue equal to or greater than 1.000. We used z-scores to select items that fit the types. For each type, we reviewed participants’ reasons for selecting statements with which they most agreed and disagreed, focusing on study participants with high factor weights. Subsequently, we used the results obtained from the analysis to interpret the characteristics by type.

## 4. Study Results

### 4.1. Results Analysis

After conducting the analysis, we extracted four types of perceptions of counselling experiences. We divided these types into four groups based on the participants’ similar subjective perceptions of ‘counselling experiences at psychological counselling centres’. Twelve participants belonged to Type 1, four to Type 2, eight to Type 3, and seven to Type 4; these types shared similar subjective perceptions about counselling experiences. The Q factor analysis results can be found in Table 4, and the eigenvalues were 10.2091 (Type 1), 2.5365 (Type 2), 2.3490 (Type 3), and 1.8618 (Type 4). The cumulative variance was 0.5185.

Correlation expresses the degree of similarity between types. According to Table 5, the correlation coefficient values are 0.322 between Types 1 and 2, 0.559 between Types 1 and 3, 0.544 between Types 1 and 4, 0.368 between Types 2 and 3, 0.405 between Types 2 and 4, and 0.460 between Types 3 and 4.

Table 6 shows the study participants and their corresponding factor weights. Based on P samples with a factor weight equal to or greater than 1.0, the higher the factor weight of a P sample, the more representative the P sample by type [29]. In Type 1 (N = 12), P25 exhibited the highest factor weight value of 2.0062; in Type 2 (N = 4), P29 had the highest factor weight of 2.0676; in Type 3 (N = 8), P4 had the highest factor weight of 1.6944. In Type 4 (N = 7), P9 had the highest, factor weight of 1.7027.

### 4.2. Analysis of the Types of Perceptions of Counselling Experiences

#### 4.2.1. Type 1: Therapeutic Counselling Relationship Valuing

We analysed the reasons for agreement and disagreement with Q statements among Type 1 participants who have high factor weights. The analysis revealed that this type aims to pursue self-insight and development, while also seeking life direction through a trusting relationship between a counsellor and a client at a professional counselling institution. Based on the analysis, we named this type the ‘therapeutic counselling relationship valuing type’. This type strongly agreed with Q1, ‘I visit a psychological counselling centre for self-understanding’ (z = 1.89), and strongly disagreed with Q26, ‘I am reluctant to visit a counselling office because I have to open my heart there’ (z = −1.77).

Participant P25, with the highest factor weight (2.0062) among Type 1 participants, made a meaningful statement:

“I visit the counselling office hoping to solve my psychological problems. Because my choice of a counsellor is based on trust, I believe there is no counselling with no counselling effect. Because psychological counselling is premised upon self-opening, I believe that counselling cannot be conducted with those who are unwilling to open their heart. Even if I paid all the counselling fees, I would quit counselling if there were no discernible counselling effect”.

The opinion reflects the type of persons who attempt to address issues through self-disclosure within a therapeutic counselling relationship where they have trust in the counsellor. They perceive that the therapeutic relationship holds greater significance than the cost aspect when choosing a counselling intervention. Table 7 summarises the statements and standard scores of Type 1.

#### 4.2.2. Type 2: Counselling Services Valuing

Type 2 prefers counselling institutions that provide kind and courteous counsellors and kind guidance, rather than focusing on counsellors’ education or experience, to help them achieve emotional stability. This type prioritises the quality of counselling services, including the principle of confidentiality, rather than focusing on problem-solving. Thus, we named this type the ‘counselling service valuing type’.

Type 2 showed the highest level of agreement with Q19, ‘I seek assistance from a psychological counsellor when I encounter difficulties in independently solving problems in my day-to-day life’ (z = 2.00). They displayed the lowest level of agreement with Q21, ‘I visit a counselling centre hoping that a counsellor will solve my problem’ (z = −2.03). In addition, statements such as Q16, ‘Counselling centre staff and counsellors are more trustworthy when they are hospitable’ (z = 1.34), Q14, ‘I need the information shared during counselling sessions to be maintained with utmost confidentiality’ (z = 1.31), and Q27, ‘I seek a counsellor who is kind and pleasant’ (z = 1.28) well represent the attributes of this type.

In particular, P29, with a factor weight of 2.067, described:
“I desire to receive better feedback when there are limitations, and I believe that psychological tests are scientific and helpful during counselling. Careers are not important for counselling, and I believe that nobody will visit a counselling centre if confidentiality is not guaranteed at the counselling centre”.
P31 (factor weight 1.8483) stated:

“I seek help from a counsellor when I cannot solve a problem on my own. The effect of counselling is not temporary but long-term, provide me with significant assistance and emotional stability. However, I do not believe counsellors can solve all problems”.

Therefore, Type 2 individuals value the service aspect during counselling activities. Table 8 summarises the statements and standard scores of Type 2.

#### 4.2.3. Type 3: Counsellor Searching

We named Type 3 the ‘counsellor searching type’. This type of client seeks counselling, prefers counsellors and counselling institutions that treat them kindly, and desires to proceed with counselling with a counsellor who fits them. Clients belonging to this type mainly show strong agreement with Q19, ‘I seek assistance from a psychological counsellor when I face difficulties in independently solving problems in my day-to-day life’ (z = 2.08), Q27, ‘I seek a counsellor who is kind and pleasant’ (z = 1.46), Q21, ‘I visit a counselling centre hoping that a counsellor will solve my problem’ (z = 1.22), Q28, ‘I would like to engage with many counsellors before choosing the ultimate one’ (z = 1.22), and Q16, ‘Counselling centre staff and counsellors are more trustworthy when they are hospitable’ (z = 1.06).

In contrast, the statements with which Type 3 clients disagree the most include Q8, ‘Psychological testing is not scientific’ (z = −2.24), and Q33, ‘I am reluctant to seek counselling because I fear that I will become dependent on it once I start’ (z = −1.92). Such opinions reflect that Type 3 clients seek assistance from a psychological counsellor when it is challenging to solve problems independently, they expect the counsellor to solve their problems, and they feel that the counsellor’s attitude greatly affects their satisfaction with counselling. Thus, this type considers the counsellor assignment process necessary during their counselling. Table 9 illustrates the statements and standard scores for Type 3.

#### 4.2.4. Type 4: Counselling Structure Valuing

Type 4 is the ‘counselling structure valuing type’. This type reflects the characteristics of valuing the professionalism of counsellors and counselling institutions, such as confidentiality, counsellor qualifications, improved emotional functions in counselling effects, and increased efficiency of counselling costs. In particular, this type shows the strongest agreement with Q 14, ‘I want the contents of counselling to be kept thoroughly confidential’ (z = 2.43). Moreover, it indicates a high level of agreement with Q6, ‘I prefer centres where the professionalism of counsellors is high’ (z = 2.09). Conversely, this type disagrees with Q31, ‘I want to be unconditionally supported at least during counselling’ (z = −1.79), and Q33, ‘I am reluctant to seek counselling because I feel like I will become dependent on counselling once I start’ (z = −1.60).

P25, with the highest factor weight (2.0062), said:

“Confidentiality must be kept, and it is vital to select a centre that maintains high professionalism among its counsellors—the higher the level of professionalism, the better the healing and recovery through counselling. If I can understand myself and improve my relationships through counselling, I will not be reluctant to rely on counselling”.

Type 4’s statements demonstrate that these clients prefer counselling institutions that observe the basic principles of psychological counselling and a counselling process conducted by professional counsellors. In addition, this type shows the characteristics of considering counselling costs for continuous counselling and seeking to receive solid counselling rather than unconditional support of simple expectations concerning counselling. Table 10 summarises Type 4’s statements and z-scores.

### 4.3. Consensus Items

We discovered eight consensus items that commonly appear in individual types (Table 11). The item that seemed to be the most positive is Q17, ‘Psychological counselling enhances my emotional stability’ (z = 1.16), and the most negative common item is Q13, ‘Even if there is no effect during counselling, I continue counselling because of the cost already paid’ (z = −0.93). Therefore, in counselling experiences, emotional stability is an essential element required by clients, and clients do not want to continue counselling without counselling effects, regardless of the cost.

## 5. Discussion and Conclusions

### 5.1. Discussion

This study applied the Q methodology to explore various perceptions of clients who experienced counselling at a psychological counselling centre. The study discovered four types of clients’ perception: (1) therapeutic counselling relationship valuing, (2) counselling services valuing, (3) counsellor searching type, and (4) counselling structure valuing.

Type 1 clients, the ‘therapeutic counselling relationship valuing type’, value self-understanding and growth through their relationship with the counsellor and set the preceding as their goal of counselling. According to a study by Timulak and Keogh [17], clients regard self-understanding and acceptance as the most important elements during counselling. This finding aligns with the tendency of Type 1 clients. Knox [30] reports that self-awareness and growth are crucial in psychotherapy and that it is possible when a therapeutic relationship with the counsellor is well formed. Such results highlight the tendency of Type 1 clients to seek to enhance their self-understanding through counselling and value their therapeutic relationship with the counsellor. According to Elliott and James [16], the deeper the client’s self-understanding, the greater the effect of counselling. Another study [31] supports the trend, suggesting that the psychological counselling community has recently tried developing a counselling model to increase self-understanding.

The second type of client, the ‘counselling services valuing type’, aims to gain emotional stability through the services of kind counsellors and counselling institutions rather than solving all problems through counselling. According to Anderson et al. [18], counselling effectiveness increases when the counsellor meets the client’s expectations and demonstrates strong counselling skills. This finding emphasises that Type 2 clients value precise services such as the counsellor’s kindness and confidentiality of the counselling institution. Similarly, Lynch [32] reports that caring for clients’ emotional needs results in client changes. These studies indicate that emotional support and kind treatment, which Type 2 clients aim to receive during counselling, can reinforce counselling effectiveness.

Regarding clients of the ‘counsellor searching type’, Type 3, the performance of psychological counselling may vary depending on clients’ perceptions of the counsellor; therefore, counsellor assignment is a critical consideration in psychological counselling centres. LaCrose [33] reported that clients’ perceptions of the counsellor greatly affected counselling results and that the positive perception of the counsellor was directly related to the effectiveness of counselling. Moreover, Harris et al. [34] propose that how the client perceives the counsellor is critical in counselling performance. Timulak and Keogh [17] emphasise that clients’ perceptions of the counsellor are essential in counselling performance. Therefore, psychological counselling centres should consider assigning appropriate counsellors to Type 3 clients.

In addition, Type 3 clients seem to want counsellors to solve their problems. This indicates that Type 3 clients seek professional counsellors who can provide effective counselling. However, it is difficult for counsellors to receive direct client feedback on their interventions; thus, it is vital to notice the client’s feedback during counselling conversations. For these Type 3 clients, counsellors require the skill to sensitively perceive the client’s behaviour and reactions [35].

For Type 4 clients, the ‘counselling structure valuing type’, emphasise the basic principles and professionalism that counsellors and institutions provide to achieve counselling goals and smooth progress. According to Fuller and Hill [36], counselling effectiveness increases when counsellors accurately explain the purpose of counselling and provide sufficient information. Hatcher [37] indicates that the decline in counselling professionalism and quality is a significant factor that causes difficulties in counselling relationships. Further, Lazar [38] states that counselling costs play an essential role in clients’ satisfaction with and maintenance of counselling and suggests that economic aspects enhance counselling efficiency. Moreover, Holahan and Slaikeu [39] emphasise that well-maintained confidentiality, an ethical aspect of counselling, will enable clients to engage more actively in counselling, thereby enhancing the smooth and effective progress of counselling.

### 5.2. Conclusions

Based on this study’s findings, we can make the following conclusions. First, counselling centres should provide clients with information to help them choose the best-fit psychological counselling centre, including options. According to Moore and Kenning [40], understanding the psychological mechanisms that underlie counselling is essential. Clients’ intentions and beliefs regarding counselling also significantly influence their decision-making process. In addition, Mitchell et al. [41] report that providing information that is focused on personal effectiveness is important when clients are selecting counselling centres. Moreover, they underscore that the information provided by counselling centres should reflect the client’s individual needs and perceptions.

Second, accurately understanding and identifying the type of client is crucial for improving the effectiveness of counselling and enhancing the counsellor’s professionalism. According to Rice et al. [42], clients felt that the effects of counselling were greater when they met a counsellor whose disposition and techniques fit them well. Leppma and Young [43] demonstrate that applying appropriate counselling techniques can effectively develop a counsellor’s professionalism, such as self-efficacy through counselling effects like reducing the client’s psychological pain and anxiety and increasing empathy. Heinonen and Nissen–Lie [44] also concluded that identifying the counsellor’s factors for counselling effects improves professionalism significantly. This finding emphasises that customised counselling approaches according to the types of clients are essential in strengthening a counsellor’s capabilities while enhancing the quality of care provided to the clients.

Third, counselling institutions must seek ways to optimise the client and counsellor assignment processes. From the intake interview to the actual counselling, psychological counselling centres undergo several complex steps, including the client’s application for counselling, counsellor assignment, training, diagnosis of case severity, and psychological testing [45]. By effectively managing these procedures, one can quickly and accurately assign an appropriate counsellor to the client. For instance, Prince [46] highlights the problem that most counselling centres provide various counselling services, such as intake interviews, crisis, individual, and group counselling, but fail to adequately tailor these services to clients’ individual characteristics. This finding emphasises that customised counselling approaches based on the types of client perceptions are essential in counselling institutions. Further, it supports the significance of the systemised approach that is proposed in this study.

In summary, this study emphasises the importance of providing comprehensive client information, understanding and identifying client types, and optimising counsellor assignments to enhance counselling effectiveness and professionalism. Tailored approaches that address the unique needs of clients and counsellors are essential for improving the overall quality of psychological counselling services.

### 5.3. Limitations

Despite this study’s valuable insights, it has certain limitations. First, being a cross-sectional study conducted at a specific time, it fails to capture any potential changes in clients’ perceptions over time. Clients’ counselling experiences and perceptions can evolve; therefore, longitudinal studies are necessary to capture these changes. Such studies would yield a more precise understanding of the evolution of client perceptions and the sustainability of counselling effects. Second, although Q methodology is beneficial for eliciting subjective perception structures, it does not offer empirical evidence regarding the connection between these structures relate and actual counselling outcomes. Therefore, further empirical research is necessary to validate the effectiveness of approaches that are based on clients’ subjective perceptions. This will provide support for counsellors and counselling institutions in adopting these methods. Third, including only 31 P sample participants limits the generalizability of the study’s findings. Future research involving larger and more diverse samples could offer more robust insights. Additionally, although including adolescents aims to capture a broader range of experiences, the limited number of adolescent participants may not adequately reflect their perspectives. Further extensive studies involving more adolescents are necessary to understand their counselling experiences thoroughly.

### 5.4. Recommendations for Future Research

These limitations must be addressed through diverse perspectives and longitudinal study designs. First, long-term longitudinal studies are necessary to accurately capture changes in clients’ perceptions and the sustainability of counselling effects. This would reflect the evolution of clients’ counselling experiences over time.

Second, although Q methodology effectively elicits subjective perception structures, empirical evidence regarding the correlation between these structures relate and actual counselling outcomes is lacking. Future research should examine this relationship to support the development of counselling strategies that effectively incorporate clients’ subjective perceptions.

Third, research involving larger and more diverse samples is needed to enhance generalizability of the findings obtained through quantitative study. Incorporating more adolescents in future studies would help to comprehensively capture the complete spectrum of their counselling experiences.

Lastly, studies that consider diverse cultural backgrounds are necessary. Exploring how cultural differences impact counselling experiences and perceptions, as well as investigating the use of digital solutions, such as virtual counselling sessions, is essential. This will help us understand the changes in counselling practices resulting from digital technology advancements. Therefore, future research should include longitudinal study designs, empirical examinations, larger and more diverse samples by quantitative study, culturally sensitive approaches, and the use of digital solutions. These efforts will improve the quality of psychological counselling and meet clients’ diverse needs.

## Figures and Tables

**Figure 1 behavsci-14-00586-f001:**
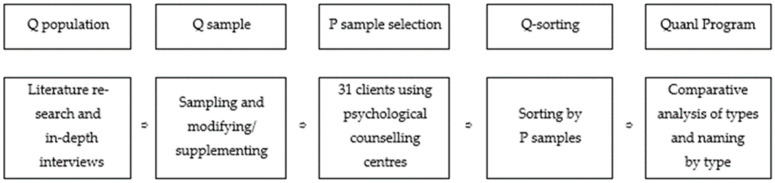
Research procedure.

**Figure 2 behavsci-14-00586-f002:**
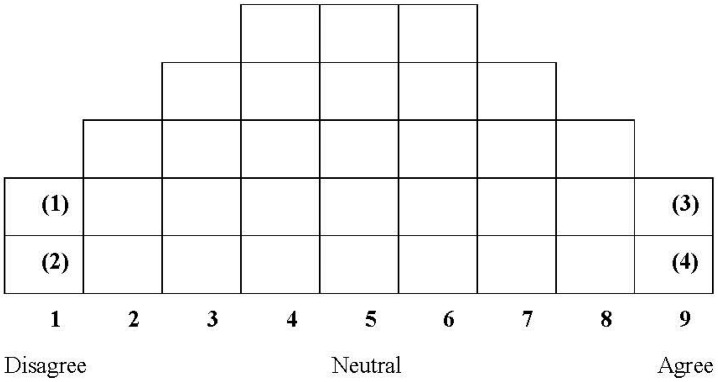
Data distribution diagram.

**Table 1 behavsci-14-00586-t001:** Q statements.

No.	Statement
Q1	I visit a psychological counselling centre for self-understanding.
Q2	The cost of the initial psychological test is exorbitant.
Q3	I do not like counselling, but I believe it is positively necessary.
Q4	My life has improved following my counselling experience.
Q5	In my opinion, the counselling sessions is overly short when they are funded by a third party.
Q6	I prefer centres that maintain a high level of professionalism among their counsellors.
Q7	There is an excessive amount of paperwork and several test processes before counselling.
Q8	Psychological testing is not scientific.
Q9	In my opinion, psychological counselling is superior to psychiatric counselling.
Q10	I seek counselling with the intention of enhancing my interpersonal relationships.
Q11	I dislike visiting a counselling centre that has outdated internal facilities.
Q12	When choosing a counselling centre, I seek recommendations from my acquaintances.
Q13	Despite the lack of observable impact during counselling, I persist with the sessions due to the financial investment already made.
Q14	I need the information discussed during the counselling session to be maintained with utmost confidentiality.
Q15	I am seeking a counsellor who will offer lifelong counselling without the need for experimentation or trial and error.
Q16	Counselling centre staff and counsellors are more trustworthy when they are hospitable.
Q17	Psychological counselling enhances my emotional stability.
Q18	I desire continuous counselling, but counselling is costly.
Q19	I desire to seek help from a psychological counsellor when I encounter difficulties in independently solving issues in my day-to-day life.
Q20	The effects of counselling are temporary.
Q21	I visit a counselling centre, with the expectation that a counsellor will solve my problem.
Q22	I prefer large-scale counselling centres, such as those with branch offices.
Q23	I select counselling centres based on their promotions. (Internet cafes, Internet searches, leaflets, etc.)
Q24	I want to select counsellors based on their education and career information.
Q25	Identifying a counselling centre can be challenging due to the varying costs across different centres.
Q26	I am reluctant to visit a counselling centre because I have to open my heart there.
Q27	I desire to meet a counsellor who is kind and pleasant.
Q28	I would like to engage with multiple counsellors before choosing the ultimate one.
Q29	I find counselling centres that resemble hospitals to be highly repulsive.
Q30	The lack of comprehensive information provided by counselling centres when I inquire is inconvenient.
Q31	I desire to be unconditionally supported, particularly during counselling.
Q32	When selecting a counselling centre, I consider factors such as accessibility, transportation convenience, and parking convenience.
Q33	I am hesitant to initiate counselling due to concerns that I may become dependent on it.

**Table 2 behavsci-14-00586-t002:** P samples.

No.	Gender	Age Group	No.	Gender	Age Group
P1	F	40s	P17	F	20s
P2	F	60s	P18	F	60s
P3	F	50s	P19	F	30s
P4	F	30s	P20	M	50s
P5	F	40s	P21	F	20s
P6	M	40s	P22	F	20s
P7	F	50s	P23	F	20s
P8	M	20s	P24	F	30s
P9	M	40s	P25	F	20s
P10	M	40s	P26	M	20s
P11	F	60s	P27	F	30s
P12	F	50s	P28	F	50s
P13	F	50s	P29	M	10s
P14	M	10s	P30	F	40s
P15	F	20s	P31	F	50s
P16	F	50s			

**Table 3 behavsci-14-00586-t003:** Sorting distribution.

Number of Statements	2	3	4	5	5	5	4	3	2
Score	−4	−3	−2	−1	0	1	2	3	4

**Table 4 behavsci-14-00586-t004:** Eigenvalues and explanatory variables in the sorting of four types.

Contents/Type	Type 1	Type 2	Type 3	Type 4
Chosen Eigenvalues	10.2091	2.5365	2.3490	1.8618
Total Variance	0.3293	0.0818	0.0758	0.0601
Cumulative	0.3293	0.4111	0.4869	0.5470

**Table 5 behavsci-14-00586-t005:** Correlations between individual types.

Contents/Type	Type 1	Type 2	Type 3	Type 4
Type 1	1.000	0.322	0.559	0.544
Type 2		1.000	0.368	0.405
Type 3			1.000	0.460
Type 4				1.000

**Table 6 behavsci-14-00586-t006:** Study participants and factor weights by types.

Study Participant	Factor Weight	Study Participant	Factor Weight
Type 1 (N = 12)		Type 2 (N = 4)	
P5	1.2783	P13	0.8442
P6	0.5030	P23	1.5888
P8	0.4803	P29	2.0676
P11	1.6701	P31	1.8483
P12	0.8873		
P15	1.4625		
P16	0.5617		
P18	1.0268		
P25	2.0062		
P26	0.7203		
P27	0.9258		
P30	0.9647		
Type 3 (N = 8)		Type 4 (N = 7)	
P1	0.7409	P2	0.9959
P4	1.6944	P3	0.5267
P14	1.3525	P7	0.6575
P17	1.1872	P9	1.7027
P19	1.0256	P10	0.7328
P22	1.1278	P20	1.0944
P24	0.4561	P21	0.4759
P28	0.4931		

**Table 7 behavsci-14-00586-t007:** Statements and standard scores of Type 1 (±1.00 or higher).

No.	Statement	Standard Score (Z-Score)
1	I visit a psychological counselling centre for self-understanding.	1.89
19	I seek assistance from a psychological counsellor when I encounter difficulties in independently solving issues in my day-to-day life.	1.52
4	My life has improved following my counselling experience.	1.35
6	I prefer centres that maintain a high level of professionalism among their counsellors.	1.27
14	I need the information shared during the counselling sessions to be maintained with utmost confidentiality.	1.24
17	Psychological counselling enhances my emotional stability.	1.11
2	The cost of the initial psychological test is overly expensive.	1.07
15	I am seeking a lifelong counsellor who will provide counselling without the need for experimentation or trial and error.	1.05
13	Despite no observable effect during counselling, I persist with the counselling due to the financial investment already made.	−1.06
29	I find counselling centres that resemble hospitals to be repulsive.	−1.19
33	I am hesitant to start counselling due to concerns that I may become dependent on it.	−1.35
20	The effects of counselling are temporary.	−1.59
23	I select counselling centres based on their promotions (Internet cafes, Internet searches, leaflets, etc.).	−1.60
26	I am reluctant to visit a counselling centre because I have to open my heart there.	−1.77

**Table 8 behavsci-14-00586-t008:** Statements and standard scores in Type 2 (±1.00 or higher).

No.	Statement	Standard Score (Z-Score)
19	I seek assistance from a psychological counsellor when I encounter difficulties in independently solving problems in my day-to-day life.	2.00
17	Psychological counselling enhances my emotional stability.	1.48
4	My life has improved following my counselling experience.	1.41
16	Counselling centre staff and counsellors are more trustworthy when they are hospitable.	1.34
14	I need the information shared during counselling sessions to be maintained with utmost confidentiality.	1.31
27	I seek a counsellor who is kind and pleasant.	1.28
22	I prefer large-scale counselling centres, such as those with branch offices.	−1.03
20	The effects of counselling are temporary.	−1.25
8	Psychological testing is not scientific.	−1.68
24	I seek to select counsellors based on their education and career information.	−1.79
1	I visit a counselling centre, hoping a counsellor will solve my problem.	−2.03

**Table 9 behavsci-14-00586-t009:** Statements and standard scores in Type 3 (±1.00 or higher).

No.	Statement	Standard Score (Z-Score)
19	I seek help from a psychological counsellor when I encounter difficulties in independently solving problems in my day-to-day life.	2.08
27	I seek a counsellor who is kind and pleasant.	1.46
21	I visit a counselling centre, hoping a counsellor will solve my problem.	1.22
28	I would like to engage with many counsellors before choosing the ultimate one.	1.22
16	Counselling centre staff and counsellors are more trustworthy when they are hospitable.	1.06
30	The lack of detailed information provided by counselling centres when I inquire is inconvenient.	−1.02
3	I do not like counselling, but I believe it is positively necessary.	−1.39
29	I find counselling centres that resemble hospitals to be repulsive.	−1.69
33	I am hesitant to initiate counselling due to concerns that I may become dependent on it.	−1.92
8	Psychological testing is not scientific.	−2.24

**Table 10 behavsci-14-00586-t010:** Statements and standard scores in Type 4 (±1.00 or higher).

No.	Statement	Standard Score (Z-Score)
14	I need the information discussed during counselling to be maintained with utmost confidentiality.	2.43
6	I prefer centres that maintain high professionalism among their counsellors.	2.09
17	Psychological counselling enhances my emotional stability.	1.39
18	I desire continuous counselling, but the counselling fee is costly.	1.33
1	I visit a psychological counselling centre for self-understanding.	1.07
21	I visit a counselling centre, hoping a counsellor will solve my problem.	−1.17
13	Despite no discernible effect during counselling, I persist with the counselling due to the financial investment already made.	−1.45
8	Psychological testing is not scientific.	−1.47
33	I am hesitant to initiate counselling due to concerns that I may become dependent on it.	−1.60
31	I desire to be unconditionally supported, at least during counselling.	−1.79

**Table 11 behavsci-14-00586-t011:** Consensus items of individual types.

No.	Statement	Standard Score (Z-Score)
17	Psychological counselling enhances my emotional stability.	1.16
16	Counselling centre staff and counsellors are more trustworthy when they are hospitable.	0.89
12	When selecting a counselling centre, I seek recommendations from my acquaintances.	0.33
7	There is an excessive amount of paperwork and several test processes before counselling.	−0.19
11	I dislike going to a counselling centre that has outdated internal facilities.	−0.29
30	The lack of detailed information provided by counselling centres when I inquire is inconvenient.	−0.44
22	I prefer large-scale counselling centres, such as those with branch offices.	−0.83
13	Despite no discernible effect during counselling, I persist with the counselling due to the financial investment already made.	−0.93

## Data Availability

Data are contained within the article; further inquiries can be directed to the corresponding author.

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
