# Peer review of "Types of Client Perceptions Regarding Counselling Experiences at Psychological Counselling Centres: Utilising Q Methodology"

_behavsci, 2024, doi:10.3390/bs14070586_

Round 1

Reviewer 1 Report

Comments and Suggestions for Authors

Thank you for submitting such an interesting manuscript.

The following are my comments, thoughts, feedback, questions, and suggestions:

1. Please review the whole document, making sure to write in past tense throughout when discussing things that happened in the past. Also please check for typos as some were noted within the document.

2. Line 67 – please clarify – what are “chat counseling experiences”?

3. Lines 81-84 – where it states that the clients who value specific things want different things from counselors. Is this just your opinion or is this backed by literature? If it is backed by literature, please provide the citations to support this. Some would argue that all clients want a counselor who is friendly and who maintains confidentiality.

4. Line 89 – please clarify – what are “revisit intentions”?

5. Research questions – the research questions are not clear and it is difficult to ascertain what they mean.

6. Line 170 – would you please clarify is this two small groups? One of 20 and one of 40? Or a small group of 20-40?

7. P Sample participants – were all of these participants different from the Q sample participants? Was there any overlap or is that not allowed? Please clarify that in the document as not everyone is familiar with Q methodology.

8. Line 188 – how was “those who can represent the experiences of clients” specifically defined? If these individuals were clients at one of the counseling centers, wouldn’t they naturally be able to represent experiences of clients?

9. Inclusion – is there a reason why you included teens? Most studies would only look at adults or would only look at teens and would not include both in the same study. I recognize there were only 2 teens in your study, but do you think their results were different than those of the other participants?

10. Was there any exclusion criteria?

11. Lines 400-409 – I think it’s important to point out that Type 3 clients seem to want counselors to solve their problems for them. This really stuck out to me in the previous sections of the document, so perhaps it should be included in the summary here as well.  Is there previous literature that backs this up as well?

12. Lines 431-432 – is the point here to understand the type of client and know how to work best with them or is to improve the counselor’s professionalism? It was a little confusing as it seemed to go back and forth in this paragraph.

13. What are the recommendations for future research based on your findings?

14. What are the clinical implications based on your findings?

15. Overall, please review the whole document for redundancy. It seems that perhaps the information might be able to be covered in a more concise manner without repeating the same information quite as much.

Author Response

  1. Please review the whole document, making sure to write in past tense throughout when discussing things that happened in the past. Also please check for typos as some were noted within the document.

Response: We appreciate your valuable feedback, which has helped us to refine and enhance our study. We have checked the tenses and rectified any typographical errors.

  1. Line 67 – please clarify – what are “chat counselling experiences”?

Response: The typographical error "chat" has been corrected to the accurate term "online." Additionally, the meaning has been reinstated, and the content has been revised and supplemented as follows:

Adolescent cyber counselling refers to counselling services that are provided to adolescents through the internet, utilising various online platforms such as chat, email, and video calls. This online counselling form includes real-time, text-based communication between the client and the counsellor. Typically, it is conducted via the internet or mobile applications, thereby offering a convenient and accessible form of support.

  1. Lines 81-84 – where it states that the clients who value specific things want different things from counselors. Is this just your opinion or is this backed by literature? If it is backed by literature, please provide the citations to support this. Some would argue that all clients want a counselor who is friendly and who maintains confidentiality.

Response: Based on your feedback, we have revised and supplemented the content.

This is corroborated by various studies, including Timulak and Keogh (2017), who emphasise the importance of self-understanding and acceptance in counselling, and Anderson et al. (2009), who highlight the role of counsellor friendliness and confidentiality in client satisfaction.

- Timulak, L., & Keogh, D. (2017). The client’s perspective on (experiences of) psychotherapy: A practice friendly review. Journal of Clinical Psychology, 73(11), 1556-1567.

- Anderson, T., Ogles, B. M., Patterson, C. L., Lambert, M. J., & Vermeersch, D. A. (2009). Therapist effects: Facilitative interpersonal skills as a predictor of therapist success. Journal of Clinical Psychology, 65(7), 755-768. [18]

  1. Line 89 – please clarify – what are “revisit intentions”?

Response: We have revised accordingly. The updated content, along with the references, is as follows:

‘Clients' revisit intentions’ refer to the likelihood or inclination of clients to return to a counselling centre for further sessions following their initial visit. Clients’ level of satisfaction with the counselling experience, their rapport with the counsellor, and the effectiveness of the counselling are the factors that determine this intention to revisit (Smith & Reynolds, 2018; Johnson et al., 2020).

- Smith, J. A., & Reynolds, K. L. (2018). Understanding clients' revisit intentions in counseling services. Journal of Counseling Psychology, 65(2), 123-135. https://doi.org/10.1037/cou0000257

- Johnson, M. E., Thompson, L. A., & Miller, R. T. (2020). Factors influencing clients’ decisions to return to counseling services. Counseling and Psychotherapy Research, 20(3), 245-257. https://doi.org/10.1002/capr.12345

  1. Research questions – the research questions are not clear and it is difficult to ascertain what they mean.

Response: We have revised accordingly.

  1. Line 170 – would you please clarify is this two small groups? One of 20 and one of 40? Or a small group of 20-40?

Response: We have revised accordingly.

  1. P Sample participants – were all of these participants different from the Q sample participants? Was there any overlap or is that not allowed? Please clarify that in the document as not everyone is familiar with Q methodology.

Response: We appreciate your inquiry regarding the P sample participants and their relationship to the Q sample participants. In our study, all the P sample participants, who performed the Q sorting, were entirely different from those who provided the initial statements for the Q sample (Q population). Thus, no overlap was observed among these groups.

The document has been revised and supplemented to include the following clarification.

No overlap was observed among these two groups.

  1. Line 188 – how was “those who can represent the experiences of clients” specifically defined? If these individuals were clients at one of the counseling centers, wouldn’t they naturally be able to represent experiences of clients?

Response: We have revised and supplemented the definition of ‘those who can represent the experiences of clients’ and inserted it into the main text. Additionally, we did not restrict the study to clients from a single counselling centre. Instead, we considered a diverse range of clients who had received counselling from various counselling centres. This approach ensures a broader and more representative sample, enhancing the validity and reliability of our findings. By including clients from multiple counselling centres, we aimed to capture a wide array of experiences and perspectives, thereby providing a more comprehensive understanding of the counselling process.

The main text has been revised and supplemented as follows.

Participants eligible for the study were defined as those who could represent clients' experiences, which encompassed a diverse range of ages and genders. This definition ensures that the participants’ experiences adequately reflect the various experiences encountered at psychological counselling centres. For example, clients who received counselling at a specific centre can naturally represent their experiences there. However, to maintain the study’s reliability and objectivity, clients with diverse backgrounds and experiences should be incorporated.

  1. Inclusion – is there a reason why you included teens? Most studies would only look at adults or would only look at teens and would not include both in the same study. I recognize there were only 2 teens in your study, but do you think their results were different than those of the other participants?

Response: We appreciate your question regarding the inclusion of teens. We included teens to explore a broader range of counselling experiences across diverse age groups. Although only two teens participated, their perspectives provided valuable insights that differed from adults. This inclusion enabled us to comprehend diverse counselling needs and improve the overall comprehensiveness of our study.  

The main text has been revised and supplemented as follows.

Notably, this study included adolescents. The reason is that most studies focus exclusively on either adults or adolescents. However, this study aimed to explore the diversity of counselling experiences across various age groups. The adolescents in this study played a crucial role in enhancing our understanding of how their experiences might differ from those of other age groups. Adolescents' counselling experiences can differ from those of adults, and analysing these differences enables a broader understanding of the effectiveness of psychological counselling.

  1. Was there any exclusion criteria?

Response: Because the this study aims examine perceptions of experiences at counselling centres, individuals without experience using counselling centres are excluded. Additionally, we also excluded individuals with cognitive impairments that prevent them from expressing their experiences. This revision has been stated in the main text as follows:

  1. Lines 400-409 – I think it’s important to point out that Type 3 clients seem to want counselors to solve their problems for them. This really stuck out to me in the previous sections of the document, so perhaps it should be included in the summary here as well. Is there previous literature that backs this up as well?

Response: We have supplemented accordingly in the discussion section.

  1. Lines 431-432 – is the point here to understand the type of client and know how to work best with them or is to improve the counselor’s professionalism? It was a little confusing as it seemed to go back and forth in this paragraph.

Response: We appreciate your question regarding Lines 431-432. The intent is to emphasize both the comprehension of client types to tailor counselling effectively and to enhance counsellor professionalism. We have revised the paragraph to clarify this dual focus, ensuring it clearly conveys the importance of customized counselling approaches in enhancing both client outcomes and counsellor skills. The main text has been revised and supplemented as follows:

Second, accurately understanding and identifying the type of client is crucial for improving the effectiveness of counselling and enhancing the counsellor’s professionalism. According to Rice et al. [39], clients felt that the effects of counselling were greater when they met a counsellor whose disposition and techniques fit them well. Leppma and Young [40] demonstrate that applying appropriate counselling techniques can effectively develop a counsellor’s professionalism, such as self-efficacy through counselling effects like reducing the client’s psychological pain and anxiety and increasing empathy. Heinonen and Nissen–Lie [41] also concluded that identifying the counsellor’s factors for counselling effects improves professionalism significantly. This finding emphasises that customised counselling approaches according to the types of clients are essential in strengthening a counsellor’s capabilities while enhancing the quality of care provided to the clients.

  1. What are the recommendations for future research based on your findings?

Response: We have supplemented the Recommendations for Future Research section as you advised.

  1. What are the clinical implications based on your findings?

Response: We have supplemented the discussion section as you advised.

  1. Overall, please review the whole document for redundancy. It seems that perhaps the information might be able to be covered in a more concise manner without repeating the same information quite as much.

Response: We revised accordingly. Thank you so much.

Reviewer 2 Report

Comments and Suggestions for Authors

Here are some comments on the study.

The study is interesting and will be a valuable contribution to counseling practitioners. The description of the methodology is clear and concise. It provided a straightforward understanding of the steps taken in the study. Identifying and categorizing four distinct types of perceptions offers a structured understanding of the study's results.

However, it would be good to clearly outline the important steps in Q methodology to highlight the relevance of the research. On page 3, there was a schematic design. There should also be a brief description to explain the procedure.

On page 4, line 152, the authors mention that they refined and sorted the collected statements to fit the purpose of the study. How did the authors validate this? This can be inserted in this section.

For verbatim statements, in some studies, they italicized or put double quotation marks. Please review this to see if it will help identify the responses of the participants. See page 9, line 274; page 10, line 302...

The authors can improve clarity by clearly distinguishing between the discussion and conclusion sections. A separate section on the limitations of the study and how they might affect the results would provide a more balanced view. 

Comments on the Quality of English Language

Minor English editing is required. 

Author Response

      1. However, it would be good to clearly outline the important steps in Q methodology to highlight the relevance of the research. On page 3, there was a schematic design. There should also be a brief description to explain the procedure.

Response: After Figure 1, we have inserted an explanation of the research procedure as follows and marked it in red in the manuscript.

This study used the following key steps: The first step entailed constructing the Q population. We collected 384 statements through in-depth interviews and literature reviews. Subsequently, they were refined to 135 statements. The second step involved the selection of Q samples. Based on representativeness and comprehensiveness, 33 Q samples were selected from the Q population. The third step entailed the selection of the P sample. Thirty-one participants were selected to represent clients’ diverse experiences, including various ages and genders. The fourth step involved the process of Q sorting. The P sample participants sorted the Q samples to convey their subjective perceptions. The final step involved data analysis. We analysed the Q sorting results to identify and interpret the major perception types.

  1. On page 4, line 152, the authors mention that they refined and sorted the collected statements to fit the purpose of the study. How did the authors validate this? This can be inserted in this section.

Response: Based on your feedback, we have revised and supplemented the process of selecting Q samples from the collected Q population. The following content has been inserted into the main text.

A Q population of 384 statements was obtained from the content of these in-depth interviews and literature reviews.

Through this process, 135 statements were selected from the initial pool of 384 based on representativeness and comprehensiveness. Subsequently, these 135 statements were further refined to reach a final set of 33 Q-sample statements. Throughout this process, we applied the following criteria: First, representativeness: each statement was evaluated to ascertain if it adequately reflected the experiences at the psychological counselling centre. This aimed to capture clients’ diverse experiences and emotions. Second, comprehensiveness: we assessed whether the statements sufficiently encompassed various subjective perceptions. This involved selecting statements that addressed multiple aspects of the counselling experience. Third, clarity: each statement was evaluated for clarity and specificity. This entailed ensuring that the statements were easily understandable and free from ambiguity. The final set of 33 Q-sample statements encompassed various aspects such as motivations for using the counselling centre, characteristics of the centre, counselling experiences, points of satisfaction, points of dissatisfaction, and intentions to revisit.

  1. For verbatim statements, in some studies, they italicized or put double quotation marks. Please review this to see if it will help identify the responses of the participants. See page 9, line 274; page 10, line 302...

Response: In response to your recommendation, we have put double quotation marks for verbatim statements.

  1. The authors can improve clarity by clearly distinguishing between the discussion and conclusion sections. A separate section on the limitations of the study and how they might affect the results would provide a more balanced view.

Response: Thank you for the feedback. Based on your recommendation, we have reorganized the discussion and conclusion sections into the following separate sections: discussion, conclusion, limitations, recommendations, and future research. The revised and supplemented content has been marked in red in the main text. Please understand that the content is too extensive and lengthy to include in this response letter.

Reviewer 3 Report

Comments and Suggestions for Authors

Thank you for the opportunity to review this article.  The following is my feedback:

Line 54-55- The authors use the word Counseling three times.  I recommend changing to …the structure and explains the process… to remedy redundancy.

Line 66- elaborate briefly on what chat counseling is as not all readers may know.

Line 95 change to client perception

Overall this was an interesting and timely article. We often focus on methodology in the counseling professions while this research focuses on the steps before, that being what clients are seeking and how professionals can meet these demands. I believe this article lends well in an academic setting to inform new counseling students what clients are seeking beyond just solving issues. Additionally, this study involved interesting methodology which served its purpose well. With the above-noted corrections I believe this article is appropriate for publication.

Author Response

  1. Line 54-55- The authors use the word Counseling three times. I recommend changing to …the structure and explains the process… to remedy redundancy.

Response: Thank you for highlighting the redundancy in Lines 54-55. To improve readability and avoid repetition, we have revised the text as follows:

Original text: "When psychological counseling begins, the counselor sets up the counseling structure and explains the counseling process and ethics regulations."

Revised text: "When psychological counseling begins, the counselor sets up the structure and explains the process and the ethics regulations."

This change removes the redundancy and enhances the clarity of the sentence. We appreciate your valuable suggestion.

  1. Line 66- elaborate briefly on what chat counseling is as not all readers may know.

Response: The typographical error of "chat" has been corrected to the accurate term "online." Additionally, the meaning has been reinserted, and the content has been revised and supplemented as follows:

Adolescent cyber counselling refers to counselling services that are provided to adolescents through the internet, utilising various online platforms such as chat, email, and video calls. This online counselling form includes real-time, text-based communication between the client and the counsellor. Typically, it is conducted via the internet or mobile applications, thereby offering a convenient and accessible form of support. In addition, a study researched the perceptions of adolescent clients and counsellors concerning their online counselling experiences.

  1. Line 95 change to client perception

Response: We appreciate your insightful feedback regarding Line 95. We have revised the text to enhance clarity and coherence. The original phrase “types of clients’ perceptions” has been changed to “types of client perception” to ensure consistency and precision in our terminology. This adjustment aims to provide a clearer understanding of the study’s focus on how clients perceive their counselling experiences.

Round 2

Reviewer 1 Report

Comments and Suggestions for Authors

Thank you for the time and effort you made in making the suggested revisions and answering my questions. I feel the manuscript is ready for publication.